# Investigation of the effectiveness of no-reference metric in image evaluation in nuclear medicine

**Shigeaki Higashiyama**[1]*, **Yutaka Katayama**[2], **Atsushi Yoshida**[1], **Nahoko Inoue**[3], **Takashi Yamanaga**[2‡], **Takao Ichida**[2‡], **Yukio Miki**[3], **Joji Kawabe**[1]

1 Department of Nuclear Medicine, Graduate School of Medicine, Osaka Metropolitan University, Osaka, Japan, 2 Department of Radiology, Osaka Metropolitan University Hospital, Osaka, Japan, 3 Department of Diagnostic and Interventional Radiology, Graduate School of Medicine, Osaka Metropolitan University, Osaka, Japan

☯ These authors contributed equally to this work.
‡ TY and TI also contributed equally to this work.
* higashiya@omu.ac.jp

**Data Availability Statement:** All relevant data are within the paper.

**Funding:** The authors received no specific funding for this work.

## Abstract

### Background

In nuclear medicine, normalized mean square error (NMSE) is widely used for image quality evaluation and machine adjustment. However, evaluating clinical images in nuclear medicine using NMSE necessitates acquiring a reference image, which is time consuming and impractical. Therefore, it is necessary to explore no-reference metrics, such as perception-based image quality evaluator (PIQE) and natural image quality evaluator (NIQE), as alternatives for evaluating the quality of clinical images used in nuclear medicine.

### Purpose

To examine whether no-reference metrics can be applied to image quality evaluations for clinical images in nuclear medicine.

### Methods

Images of the Hoffman Brain Phantom containing 18F–fluoro-2-deoxy-D-glucose (FDG) were obtained using Biograph Vision (Siemens Co., Ltd). From the collected images, 14 images with varying pixel counts and acquisition times were created. Sixteen images were visually evaluated by five image experts and ranked accordingly. Image quality was assessed using NMSE, PIQE, and NIQE, and rankings were calculated based on these scores.

### Results

The Spearman's significance test revealed a strong correlation between image quality evaluations using PIQE and visual evaluations by specialists (p<0.0001). PIQE demonstrated

**Competing interests:** The authors have declared that no competing interests exist.

comparable performance to image experts in evaluating image quality, suggesting its potential for clinical image quality assessment in nuclear medicine.

## Conclusions

PIQE offers a viable method for evaluating image quality in nuclear medicine, presenting a promising alternative to traditional visual inspection methods.

## Introduction

The advent of artificial intelligence (AI)-based image processing approaches, such as generative adversarial network (GAN)-based models, has sparked significant interest in image quality assessment [1, 2]. However, traditional full-reference metrics such as peak signal-to-noise ratio (PSNR) or structural similarity (SSIM) may not effectively evaluate images generated using GANs [3, 4]. In contrast, no-reference metrics offer a promising solution for evaluating image quality when a reference image is unavailable.

Although full-reference metrics are commonly used in medical image evaluation, particularly in nuclear medicine, their reliance on reference images limits their applicability in clinical settings [5]. Normalized mean square error (NMSE), a prevalent full-reference metric, requires a long-term captured reference image corresponding to the target image, making it impractical for clinical evaluations [6–8]. Additionally, the lack of common training and standard image data further complicates image evaluation in nuclear medicine [9, 10]. For positron emission tomography (PET) images, efforts such as quantitative imaging biomarkers and harmonization have been made to evaluate pixel values obtained from images captured using different devices as comparable indicators [11, 12]. However, these are primarily intended to use pixel values as quantitative biomarkers and do not achieve image standardization [13, 14]. To address these challenges, this study investigated the efficacy of no-reference metrics, specifically PIQE and NIQE, in evaluating image quality for clinical images in nuclear medicine [15–17]. By comparing the results of no-reference metric evaluations with visual evaluations by specialists, we demonstrated the potential of these metrics in clinical practice.

### Contributions and findings

- We proposed using no-reference metrics, namely PIQE and NIQE, for the evaluation of image quality in clinical nuclear medicine.
- We demonstrated a strong correlation between the results of PIQE and visual evaluation by specialists, indicating the potential of PIQE in clinical image quality assessment.
- We confirmed the feasibility of utilizing no-reference metrics as alternative methods for image evaluation in nuclear medicine.

## Materials and methods

### Image acquisition method and analysis

The images were obtained using a Hoffman 3D brain phantom (Data Spectrum Co., Ltd) containing 26 MBq of $^{18}$F–fluoro-2-deoxy-D-glucose (FDG) with a 3D model and 1800 s of imaging time. The Images were collected according to the protocol for brain PET imaging distributed by the Japanese Society of Nuclear Medicine and the PET Nuclear Medicine Committee [18]. Biograph Vision 450(Siemens Co., Ltd.), used for clinical examination at our hospital, was used for imaging and data collection.

For this study, one axial image slice was selected from the acquired brain phantom images, depicting the frontal/temporal lobe and bilateral ventricles of the bilateral cerebral hemisphere and basal ganglia. Such images are commonly used in the study of brain PET images using phantoms [19, 20].

For evaluation, we prepared images with seven different acquisition times: 120, 180, 300, 360, 450, 600, and 900 s. The collection matrix for a total of eight seed collection times was 440-pixels. To assess images with different pixel counts, 880-pixel images (used for clinical examinations at our facility) corresponding to each of the eight acquisition times were generated. Images with varying acquisition times and pixel counts were created and extracted using Biograph Vision 450.

The imaging and image reconstruction conditions were as follows:

- Pixel size: 0.825 × 0.825 mm

- FOV: 363 mm

- Slice thickness: 3 mm

- Reconstruction conditions: Ordinary Poisson ordered-subsets expectation maximization with point-spread function and time-of-flight modeling of 214 ps

- Random correction: Delayed coincidence measurement

- Single scatter simulation

- Subset: 5

- Iteration: 8

- Filter: all-pass

- For 440-matrix size: $0.825 \times 0.825 \times 2$ mm$^3$

- For 880-matrix size: $0.4125 \times 0.4125 \times 2$ mm$^3$

- Computed tomography attenuation correction

- 80 mAs

- 120 kV

- Slice thickness: 3 mm

- Pitch: 0.55 mm

- MATLAB (MathWorks Co., Ltd) was used to calculate the noise metric

- JMP (SAS Japan Co., Ltd) was used for statistical analysis

### Evaluator and image

Three qualified diagnostic radiologists and nuclear medicine specialists, along with four nuclear medicine technologists, were selected to visually evaluate the images. Among the four technologists, two had over 15 years of clinical experience in nuclear medicine, while the other two were inexperienced.

Fig 1 depicts an image captured at 1800 s; a total of eight images were captured with different acquisition times—120, 180, 300, 360, 450, 600, and 900 s. The pixel count of the image in

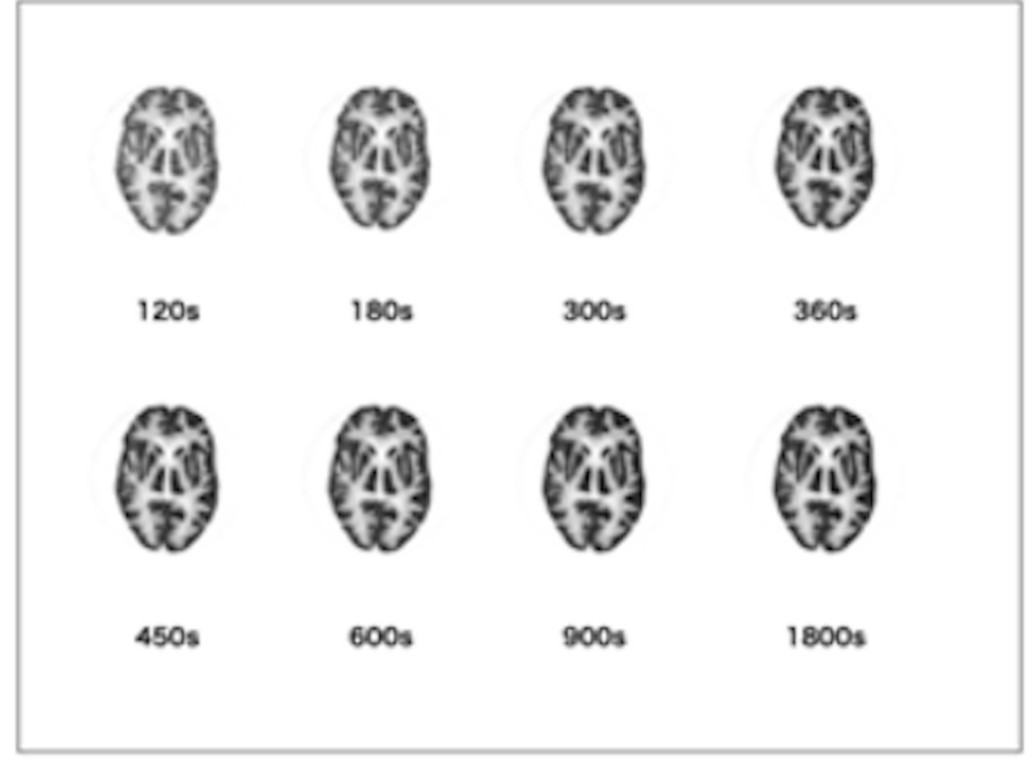

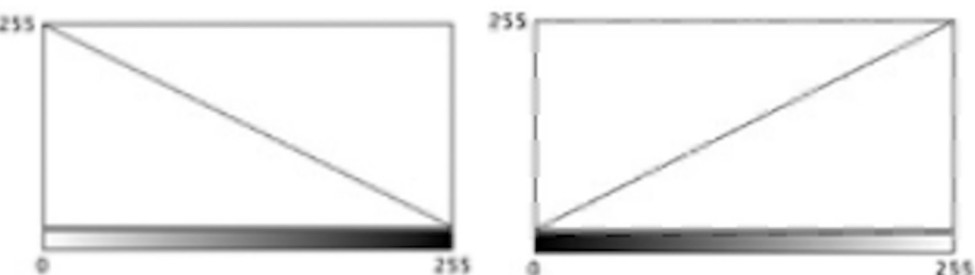

**Fig 1. A Hoffman 3D brain phantom with 26 MBq of $^{18}$F-fluoro-2-deoxy-D-glucose (FDG) with a 440-pixel matrix were obtained over 1800 s.** One slice of the axial image that depicts the frontal and temporal lobes, bilateral lateral ventricles, and basal ganglia was selected from the acquired brain phantom images. Images with different collection times (120, 180, 300, 360, 450, 600, and 900 s) were prepared with 326x188 pixels and RGB with 239K. A total of eight image types with different collection times and color scale are shown.

Fig 1 is 440 s. Fig 2 displays eight images, each containing 880 matrix size, corresponding to each acquisition time shown in Fig 1.

The following two items were defined as the visual evaluation criteria:

(1) A clear delineation of the basal ganglia limbus and its clear separation from the cerebral white and gray matter.

(2) Uniform accumulation of FDG in the basal ganglia and cerebral white matter.

Previous studies addressed image quality by quantifying "the contrast between gray-matter structures and a white matter structure" and determining "the sharpness of the gray/white-matter" [19, 20].

The aforementioned criteria were set to evaluate the sliced image used in this study. The evaluation method necessitated numerical ranking. Therefore, we employed a paired comparison method to rank the evaluations by the evaluator, referencing previous research [21].

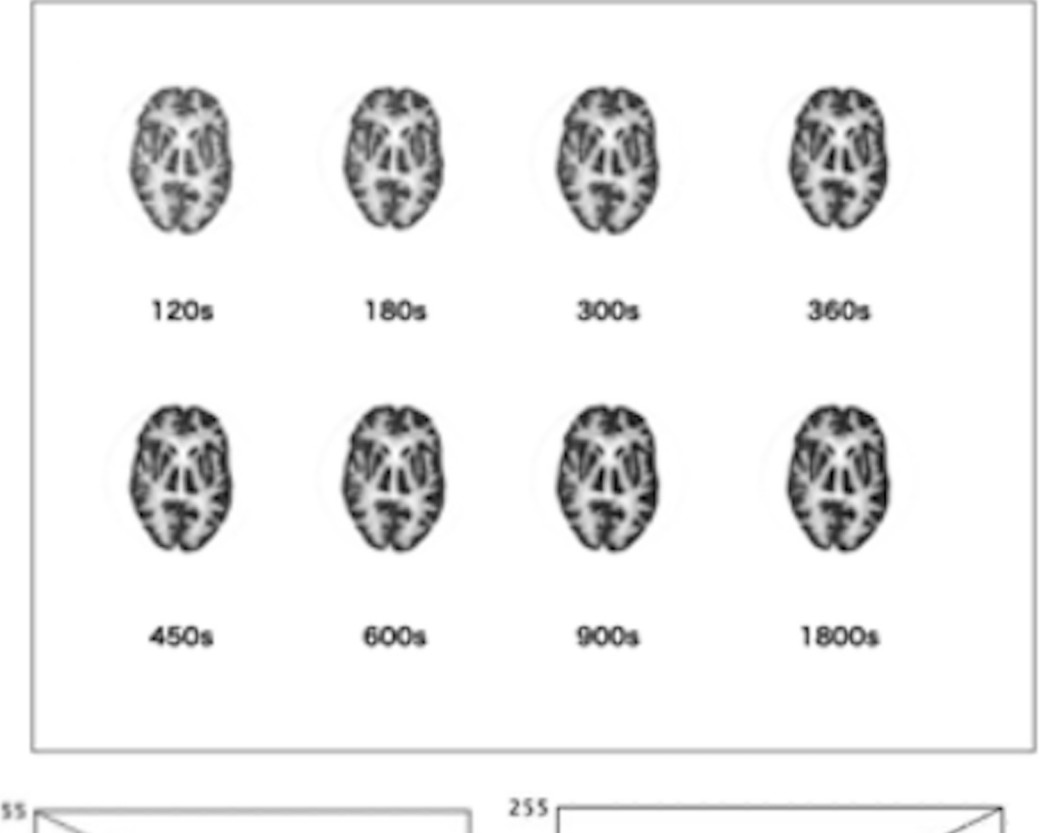

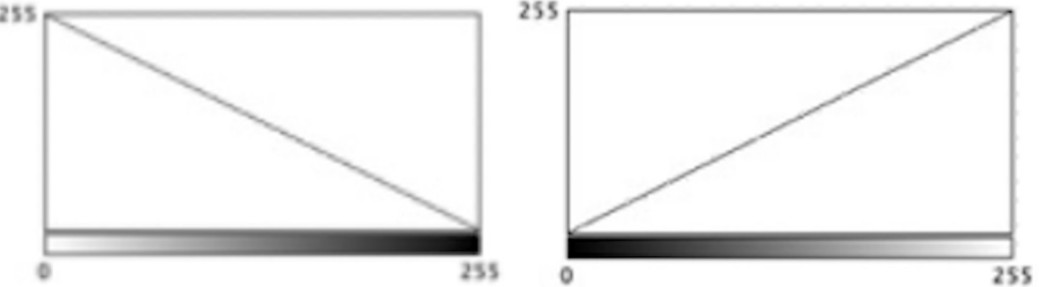

**Fig 2. 880-pixel images corresponding to each of the eight imaging times and color scale are shown.** Images with different collection times were prepared with 326x188 pixels and RGB with 239K.

## Visual evaluation method

The paired comparison method was used for visual evaluation: two images were displayed on the left and right sides of a monitor. Among the total of 16 images, two were selected to ensure that the same image was not displayed. By pairing different images on the left and right sides, a total of 240 image types were prepared and displayed randomly. The evaluator was unaware of which two images would be presented. In Fig 3, a 440-pixel image acquired at 180 s displayed on the left, and a 880-pixel image acquired at 900 s is displayed on the right. Numbers corresponding to all the 240 images are shown in Fig 2, which serves as a score entry sheet. Fig 3 corresponds to square 29 in Fig 4.

Fourteen evaluation score sheets (Fig 4) were prepared for the seven evaluators to assess items 1 and 2. The table in Fig 4 was not shown to the evaluators, who visually evaluated the 240 images displayed in a random order. If the image displayed on the right side was of better quality than that on the left side, one point was assigned to the cell in Fig 4 for that image.

**Fig 3. An image presented to evaluators.** A 440-pixel at 180 s image and a 880-pixel at 900 s image are displayed on the left and right sides, respectively. An image presented to evaluators. On the left is a 440-pixel at 180 s image, and on the right is a 880-pixel at 900 s image. This image corresponds to square number 29 shown in Fig 2.

As shown in Fig 3, if the image on the right showed more uniform accumulation of FDG in the basal ganglia and white matter, the cell in square 29 of the evaluation score sheet for item 2 was assigned a score of 1. Two images identical to Fig 3 were included in the presentation but

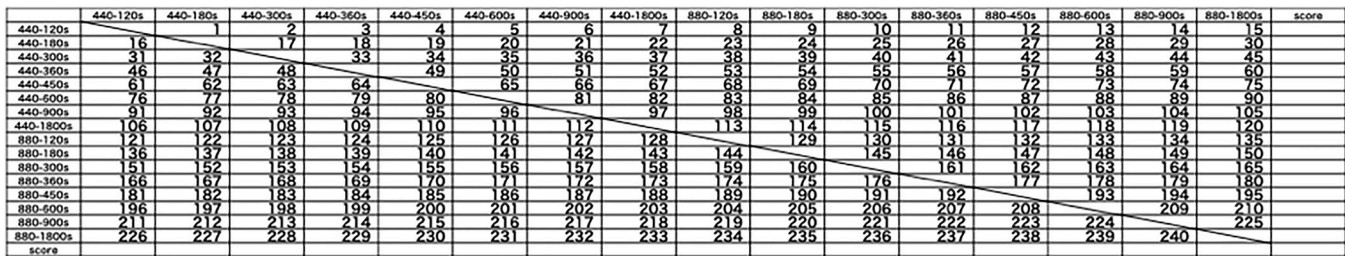

**Fig 4. Evaluation sheet for the images (out of 240 images) presented to the evaluator.** When the evaluator records the score, the cell in Fig 2 is blank, and the score is entered for the left and right images compared using the pairwise method. The rating is the total score in the leftmost column and bottom row.

arranged in opposite directions, that is, the 900 s image with 880 pixels was presented on the left side, and the 180 s image with 440 pixels was presented on the right side corresponding to square 212 in Fig 4. In this case, if the image on the left side was better, a score of 0 was assigned to square 212.

Previous reports have visually scored PET images with different acquisition times and the degree of glucose metabolism and malignancy in thyroid tumors on a 5-point scale [22, 23]. Based on these reports, we scored the images based on their acquisition times and pixel count.

When the evaluator recorded the score, it was entered for the left and right images compared using the pairwise method. For the displayed image, as shown in Fig 3, a score of 0 or 1 was recorded in the corresponding cell of Fig 4. This was performed for evaluation items 1 and 2.

Higher total scores in the rightmost column of Fig 4 represent better results. Additionally, lower total scores in the bottom row represent better results. These scores were totaled, and the average values were calculated to obtain the visual evaluation scores and ranks.

## Evaluation of NMSE

For physical evaluation, we used a physical index based on the NMSE, which has been conventionally used to calculate the similarity between reference and target images. The ideal and acquired images were used as the reference and target images, respectively [24]. NMSE normalizes the target images using the maximum number of pixels. The smaller the calculated value, the closer it is to the ideal target image [24]. The computation is as shown in Eq 1.

$$\text{NMSE} = f(x) = \frac{\sum \left(g(x,y) - f(x,y)\right)^2}{\sum f(x,y))^2} \tag{1}$$

where $f(x, y)$ refers to the reference image, and $g(x, y)$ refers to the target image.

The target image for 440-pixel images obtained with acquisition times of 120, 180, 300, 360, 450, 600, and 900 s was a 440-pixel image with an acquisition time of 1800 s. The NMSE for the images with seven other acquisition times was calculated. NMSE value was calculated for the 880-pixel images in the same manner.

## Evaluation of no-reference metric

PIQE is a no-reference perception-based image quality evaluation method for real-world images. It uses the mean subtraction contrast normalization coefficient to calculate the image quality score [15]. The natural image quality evaluator (NIQE) is an existing blind image quality evaluation method that relies on opinion-based supervised learning to predict quality scores [25]. However, PIQE is an unsupervised method that does not require a learning model [15].

PIQE is inspired by the following principles of human perception of image quality. First, human visual attention is strongly directed to prominent points in an image or spatially active areas; this property is adapted by estimating distortions only in spatially prominent areas [8]. Second, local quality at the block/patch level is the overall quality of the image that humans perceive, and this property is addressed by calculating the distortion level at the local block level of size n × n, where n = 16 [15].

Fig 5 shows a block diagram of the proposed method. The input image was preprocessed, followed by a block-level analysis to identify the distortion [15]. Each distorted block was assigned a score based on the distortion type, and the block-level scores were then pooled to determine the overall image quality. In addition to the quality score, it also generates a spatial quality map that can be effectively used in other applications.

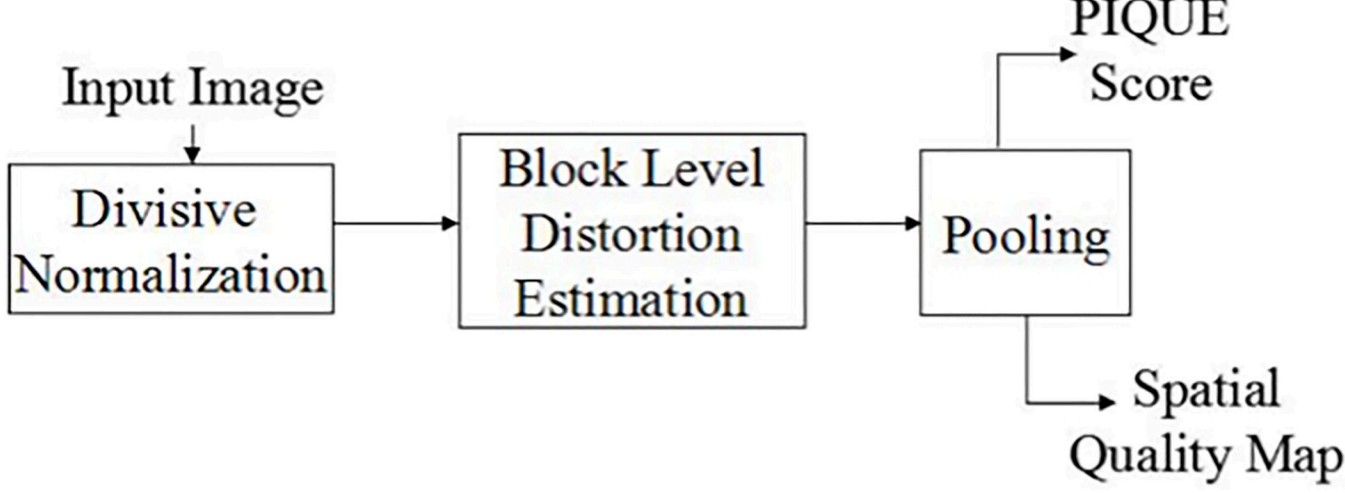

**Fig 5. Block diagram of the proposed method.** The input image was subjected to a preprocessing step. A block-level analysis was performed to identify the distortion, and each distorted block was assigned a score based on the distortion type. The block-level scores were pooled to determine the overall image quality.

In contrast, NIQE uses only measurable deviations from statistical regularities observed in natural images to calculate image quality scores in a completely blind manner [25]. It builds a collection of "quality-aware" statistical features based on a simple and successful spatial domain natural scene statistics (NSS) model [26, 27].

The distorted image quality is expressed as a simple distance metric between the model statistic and distorted image statistic [17]. Lower PIQE and NIQE scores indicate better imaging evaluations [15, 25].

No-reference metrics do not require a reference image. Therefore, the image quality was evaluated using PIQE and NIQE for both the 440-pixel and 880-pixel images obtained with eight different acquisition times: 120, 180, 300, 360, 450, 600, 900, and 1800 s.

Spearman's rank difference test was performed. It is used in studies comparing interpretation results from AI-based methods with those of experienced readers, and for the comparison between human observers and mathematical models such as the channelized Hotelling observer [28, 29]. The significance level was set at $P < 0.05$.

To demonstrate that there was no significant difference in the ranking of PIQE results, the differences in PIQW values for each rank from 1st to 16th were calculated, resulting in 13 values. These 13 data points were divided into three groups, and Mann–Whitney's U test was performed on them. The significance level was set at $P < 0.05$.

### Evaluation of uniformity

To assess uniformity, a Region of Interest (ROI) was set on each image subjected to visual evaluation, and pixel values were measured. ROIs was positioned at the medulla of the frontal, temporal, and occipital lobes, ensuring that one edge of the ROI could be measured without crossing the boundary between the cortex and medulla. Fig 6 shows the site of the ROIs setting. The arrow in Fig 6 indicate the location of the ROI in the frontal lobe, the double arrow indicates the location of the ROI in the temporal lobe, and the arrowhead indicates the location of the ROI in the occipital lobe. Referring to previous literature, the size of the ROI was set to 5 mm in diameter [30, 31].

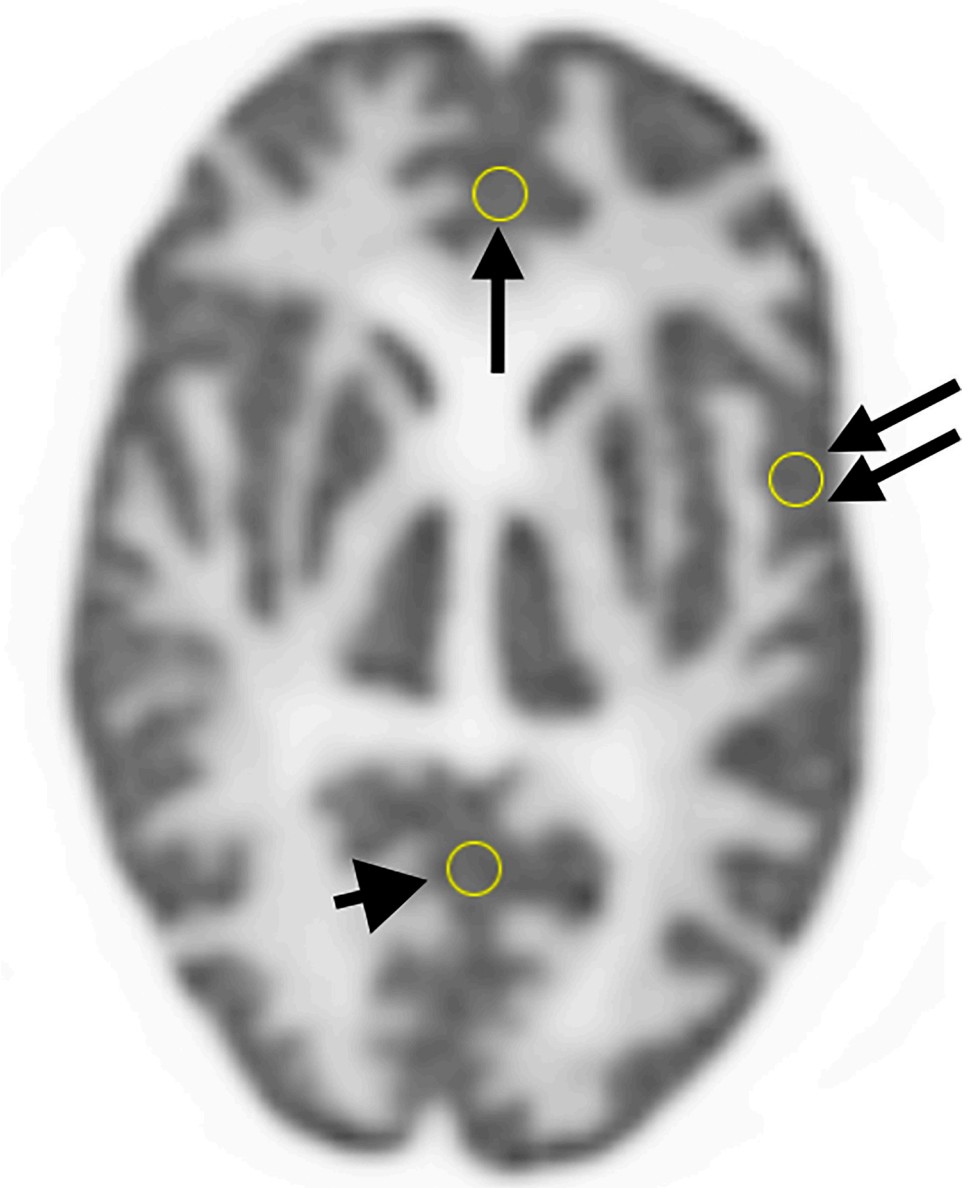

**Fig 6. Chart of region of interest (ROI) set to evaluate uniformity.** The arrow indicates the location of the ROI in the frontal lobe, the double arrow indicates the location of the ROI in the temporal lobe, and the arrowhead indicates the location of the ROI in the occipital lobe. Referring to previous literature, the size of the ROI was set to 5 mm in diameter.

Numerical evaluation was performed using the coefficient of variation (CV). CV was calculated for the images with each pixel count and acquisition time. The calculation is shown in Eq 2.

$$\mathrm{CV} = \sigma/\bar{x} \times 100, \tag{2}$$

where σ is standard deviation (SD) and $\bar{x}$ is average value.

## Ethics statement

This study exclusively utilized phantom images and did not involve the use of clinical images or human imaging data. As such, there was no requirement for approval from the Ethics Committee. Furthermore, as patient image data were not utilized, no explanation or consent was sought from any patient.

# Results

## Results of visual evaluation

Tables 1 and 2 show the scores of the two items for each image from all raters, obtained using the paired comparison method. Evaluators 6 and 7, who were inexperienced, were excluded from the analysis as their results tended to differ from those of the other evaluators.

Fig 4 shows a scoring sheet for entering 0 and 1 to indicate the superiority of images in the paired comparison method. The bottom row of Fig 4 is a column for entering the sum of these numbers vertically for each image. Table 1 represents the results of that column. Higher scores represent better evaluation results. The rightmost column of Fig 4 is a column for entering the sum of these numbers horizontally for each image. Table 2 represents the results of that column. Lower scores represent better evaluation results.

The average visual evaluation results obtained using the pairwise comparison method for all items is shown in Table 3.

In Table 3, the scores of an 880-pixel image acquired at the specified shooting time are highlighted in bold. It was observed that for both the 880-pixel and 440-pixel images, higher scores were achieved with longer shooting times. Additionally, for images with acquisition times other than 120 s, a higher score was obtained when the number of pixels was 880.

**Table 1. Scores for items 1 and 2 from seven evaluators.**

|  | Evaluator | 1 | 1 | 2 | 2 | 3 | 3 | 4 | 4 | 5 | 5 | 6 | 6 | 7 | 7 |
|---|---|---|---|---|---|---|---|---|---|---|---|---|---|---|---|
|  | **Evaluation Item** | **1** | **2** | **1** | **2** | **1** | **2** | **1** | **2** | **1** | **2** | **1** | **2** | **1** | **2** |
| Target |  |  |  |  |  |  |  |  |  |  |  |  |  |  |  |
| 440-120s | Score | 0 | 0 | 1 | 1 | 0 | 0 | 1 | 1 | 1 | 1 | 1 | 1 | 1 | 1 |
| 880-120s | Score | 1 | 1 | 1 | 1 | 0 | 0 | 0 | 0 | 0 | 0 | 0 | 0 | 0 | 0 |
| 440-180s | Score | 2 | 2 | 0 | 1 | 2 | 2 | 2 | 2 | 2 | 2 | 1 | 3 | 2 | 2 |
| 880-180s | Score | 3 | 3 | 3 | 3 | 3 | 3 | 3 | 3 | 3 | 3 | 2 | 2 | 3 | 3 |
| 440-300s | Score | 4 | 4 | 4 | 6 | 5 | 5 | 4 | 4 | 4 | 4 | 4 | 4 | 5 | 5 |
| 880-300s | Score | 5 | 5 | 5 | 5 | 5 | 5 | 5 | 5 | 5 | 5 | 4 | 4 | 5 | 5 |
| 440-360s | Score | 9 | 9 | 6 | 6 | 7 | 7 | 6 | 6 | 6 | 6 | 7 | 7 | 9 | 8 |
| 880-360s | Score | 8 | 8 | 8 | 8 | 6 | 6 | 7 | 7 | 7 | 7 | 4 | 6 | 8 | 7 |
| 440-450s | Score | 7 | 7 | 6 | 6 | 9 | 9 | 8 | 8 | 8 | 8 | 7 | 8 | 9 | 9 |
| 880-450s | Score | 8 | 8 | 9 | 9 | 8 | 8 | 9 | 9 | 9 | 9 | 7 | 9 | 10 | 10 |
| 440-600s | Score | 10 | 10 | 7 | 8 | 11 | 11 | 10 | 10 | 11 | 11 | 10 | 10 | 11 | 10 |
| 880-600s | Score | 12 | 12 | 11 | 11 | 10 | 11 | 11 | 11 | 11 | 11 | 10 | 11 | 11 | 11 |
| 440-900s | Score | 11 | 11 | 10 | 11 | 13 | 13 | 12 | 12 | 12 | 12 | 12 | 12 | 13 | 13 |
| 880-900s | Score | 12 | 12 | 13 | 13 | 13 | 13 | 12 | 12 | 13 | 13 | 13 | 13 | 12 | 13 |
| 440-1800s | Score | 14 | 14 | 12 | 13 | 15 | 15 | 13 | 13 | 13 | 13 | 13 | 15 | 15 | 15 |
| 880-1800s | Score | 15 | 15 | 15 | 15 | 15 | 15 | 15 | 15 | 14 | 14 | 14 | 15 | 15 | 15 |

**Table 2. Scores for items 1 and 2 from seven evaluators.**

| | Evaluator | 1 | 1 | 2 | 2 | 3 | 3 | 4 | 4 | 5 | 5 | 6 | 6 | 7 | 7 |
|---|---|---|---|---|---|---|---|---|---|---|---|---|---|---|---|
| | Evaluation Item | 1 | 2 | 1 | 2 | 1 | 2 | 1 | 2 | 1 | 2 | 1 | 2 | 1 | 2 |
| Target | | | | | | | | | | | | | | | |
| 440-120s | Score | 15 | 15 | 14 | 14 | 14 | 14 | 14 | 14 | 14 | 14 | 13 | 14 | 14 | 13 |
| 880-120s | Score | 14 | 14 | 13 | 14 | 14 | 14 | 15 | 15 | 15 | 15 | 15 | 15 | 15 | 15 |
| 440-180s | Score | 13 | 13 | 14 | 14 | 13 | 13 | 13 | 13 | 13 | 13 | 12 | 12 | 12 | 13 |
| 880-180s | Score | 12 | 12 | 12 | 12 | 12 | 12 | 12 | 12 | 12 | 12 | 12 | 13 | 12 | 13 |
| 440-300s | Score | 11 | 11 | 11 | 11 | 11 | 11 | 11 | 11 | 11 | 11 | 9 | 10 | 10 | 11 |
| 880-300s | Score | 10 | 10 | 6 | 6 | 11 | 11 | 10 | 10 | 10 | 10 | 9 | 10 | 10 | 10 |
| 440-360s | Score | 8 | 8 | 11 | 9 | 8 | 8 | 9 | 9 | 9 | 9 | 6 | 7 | 9 | 7 |
| 880-360s | Score | 7 | 7 | 7 | 7 | 9 | 9 | 8 | 8 | 8 | 8 | 7 | 9 | 9 | 8 |
| 440-450s | Score | 8 | 7 | 7 | 8 | 6 | 6 | 7 | 7 | 7 | 7 | 7 | 7 | 8 | 8 |
| 880-450s | Score | 7 | 7 | 7 | 5 | 7 | 7 | 6 | 6 | 6 | 6 | 7 | 7 | 9 | 8 |
| 440-600s | Score | 8 | 5 | 5 | 5 | 5 | 5 | 5 | 5 | 5 | 5 | 4 | 5 | 6 | 7 |
| 880-600s | Score | 3 | 3 | 3 | 4 | 5 | 5 | 3 | 3 | 4 | 4 | 4 | 4 | 5 | 5 |
| 440-900s | Score | 4 | 4 | 3 | 3 | 3 | 3 | 2 | 2 | 3 | 3 | 1 | 3 | 3 | 3 |
| 880-900s | Score | 3 | 3 | 1 | 2 | 3 | 3 | 2 | 2 | 2 | 2 | 1 | 2 | 3 | 3 |
| 440-1800s | Score | 1 | 1 | 1 | 1 | 1 | 1 | 1 | 1 | 0 | 0 | 1 | 1 | 1 | 1 |
| 880-1800s | Score | 0 | 0 | 0 | 0 | 1 | 1 | 0 | 0 | 0 | 0 | 0 | 1 | 1 | 1 |

## Results of evaluation using NMSE

The target image for the 440-pixel and 880-pixel images obtained with acquisition times of 120, 180, 300, 360, 450, 600, and 900 s were images with an acquisition time of 1800 s and 440- and 880-pixels in size, respectively. The evaluation values for the images with seven other acquisition times were calculated using the NMSE. The results are shown in Tables 4 and 5.

**Table 3. Average scores for each image and their ranking from the five experienced evaluators.**

| Target image | Average scores in descending order | Average scores in ascending order | Total ranks |
|---|---|---|---|
| **880-1800s** | **14.8** | **0.2** | 1 |
| 440-1800s | 13.5 | 0.8 | 2 |
| **880-900s** | **12.6** | **2.3** | 3 |
| 440-900s | 11.7 | 3 | 4 |
| **880-600s** | **11.1** | **3.7** | 5 |
| 440-600s | 9.9 | 5.3 | 6 |
| **880-450s** | **8.6** | **6.4** | 7 |
| 440-450s | 7.6 | 7.1 | 8 |
| **880-360s** | **7.2** | **7.8** | 9 |
| 440-360s | 6.8 | 8.8 | 10 |
| **880-300s** | **5** | **9.4** | 11 |
| 440-300s | 4.4 | 11 | 12 |
| **880-180s** | **3** | **12** | 13 |
| 440-180s | 1.7 | 13.2 | 14 |
| **880-120s** | **0.4** | **14.3** | 16 |
| 440-120s | 0.6 | 14.2 | 15 |

The results for the 880-pixel images is shown in bold.

**Table 4. NMSE scores for seven images with 440 pixels.**

| Target image | NMSE score | Rank |
|---|---|---|
| 440-900s | 0.008389 | 1 |
| 440-600s | 0.014692 | 2 |
| 440-450s | 0.015165 | 3 |
| 440-360s | 0.018685 | 4 |
| 440-300s | 0.027426 | 5 |
| 440-180s | 0.036682 | 6 |
| 440-120s | 0.052736 | 7 |

NMSE: Normalized mean square error

Table 6 summarizes all the results and arranges them in the order of the NMSE score. For most images, the NMSE value improved and approached 0 as the acquisition time increased. Additionally, the physical evaluation results of images with 880 pixels were better than those of images with 440-pixels.

## Results of evaluation using PIQE and NIQE

The results of the physical evaluation using PIQE are presented in Table 7. Images with a lower no-reference metric value, longer acquisition time, and 880 pixels showed better results. Spearman's significance test of the visual evaluation results and PIQE rankings showed a rank correlation coefficient (rs) of 0.9559 (p < 0.0001), indicating a strong correlation between the two methods (Fig 7).

Lower scores represent better image quality.

Fig 8 shows the results of Mann–Whitney's U test, where PIQE differences were divided into three groups based on numerical rankings. The rankings of PIQE were classified into three groups: group 1 consisted of the differences between 1st and 4th place, comprising three numbers; group 2 included the differences between 5th and 12th place, comprising seven numbers; and group 3 comprised the values from 13th to 16th place. No significant difference was observed among these groups. The P value, which is the test value for groups 1 and 2, was p = 0.3619, the P value for groups 2 and 3 was p = 0.175, and the P value for groups 1 and 3 was p = 0.833.

Results of the physical evaluation using NIQE are also shown in Table 8. Spearman's significance test of the visual evaluation and NIQE rankings yielded a rs of 0.2324 (p = 0.3865), indicating no significant correlation between the two methods (Fig 9).

**Table 5. NMSE scores for seven images with 880 pixels.**

| Target image | NMSE score | Rank |
|---|---|---|
| 880-900s | 0.007257 | 1 |
| 880-600s | 0.012808 | 2 |
| 880-450s | 0.01455 | 3 |
| 880-360s | 0.018592 | 4 |
| 880-300s | 0.0262 | 5 |
| 880-180s | 0.035974 | 6 |
| 880-120s | 0.050524 | 7 |

NMSE: Normalized mean square error

**Table 6. NMSE scores for all 440- and 880-pixel images except for the 1800 s images and their ranking.**

| Target image | NMSE score | Rank |
|---|---|---|
| 880-900s | 0.007257 | 1 |
| 440-900s | 0.008389 | 2 |
| 880-600s | 0.012808 | 3 |
| 440-600s | 0.014692 | 5 |
| 880-450s | 0.01455 | 4 |
| 440-450s | 0.015165 | 6 |
| 880-360s | 0.018592 | 7 |
| 440-360s | 0.018685 | 8 |
| 880-300s | 0.0262 | 9 |
| 440-300s | 0.027426 | 10 |
| 880-180s | 0.035974 | 11 |
| 440-180s | 0.036682 | 12 |
| 880-120s | 0.050524 | 13 |
| 440-120s | 0.052736 | 14 |

NMSE: Normalized mean square error

## Results of the uniformity evaluation

Tables 9–11 shows the numerical results of the uniformity rating for the three areas: the frontal lobe, the temporal lobe, and the occipital lobe. Figs 10–12 is a graph of the numerical results of Tables 9–11.

## Discussion

In this study, we examined whether no-reference metrics can be applied for the quality evaluation of clinical images in nuclear medicine. The visual assessment of the images by five raters

**Table 7. PIQE scores and their ranking.**

| Target image | PIQE score | Rank |
|---|---|---|
| 880-1800s | 60.3786 | 1 |
| 440-1800s | 60.7609 | 2 |
| 880-900s | 62.7803 | 3 |
| 440-900s | 65.3957 | 4 |
| 880-600s | 66.0972 | 5 |
| 440-600s | 69.7924 | 9 |
| 880-450s | 67.1711 | 6 |
| 440-450s | 72.1406 | 10 |
| 880-360s | 68.1758 | 7 |
| 440-360s | 72.5463 | 11 |
| 880-300s | 68.1903 | 8 |
| 440-300s | 72.7048 | 12 |
| 880-180s | 74.1163 | 13 |
| 440-180s | 75.5763 | 14 |
| 880-120s | 77.4794 | 15 |
| 440-120s | 79.5117 | 16 |

PIQE: Perception-based image quality evaluator

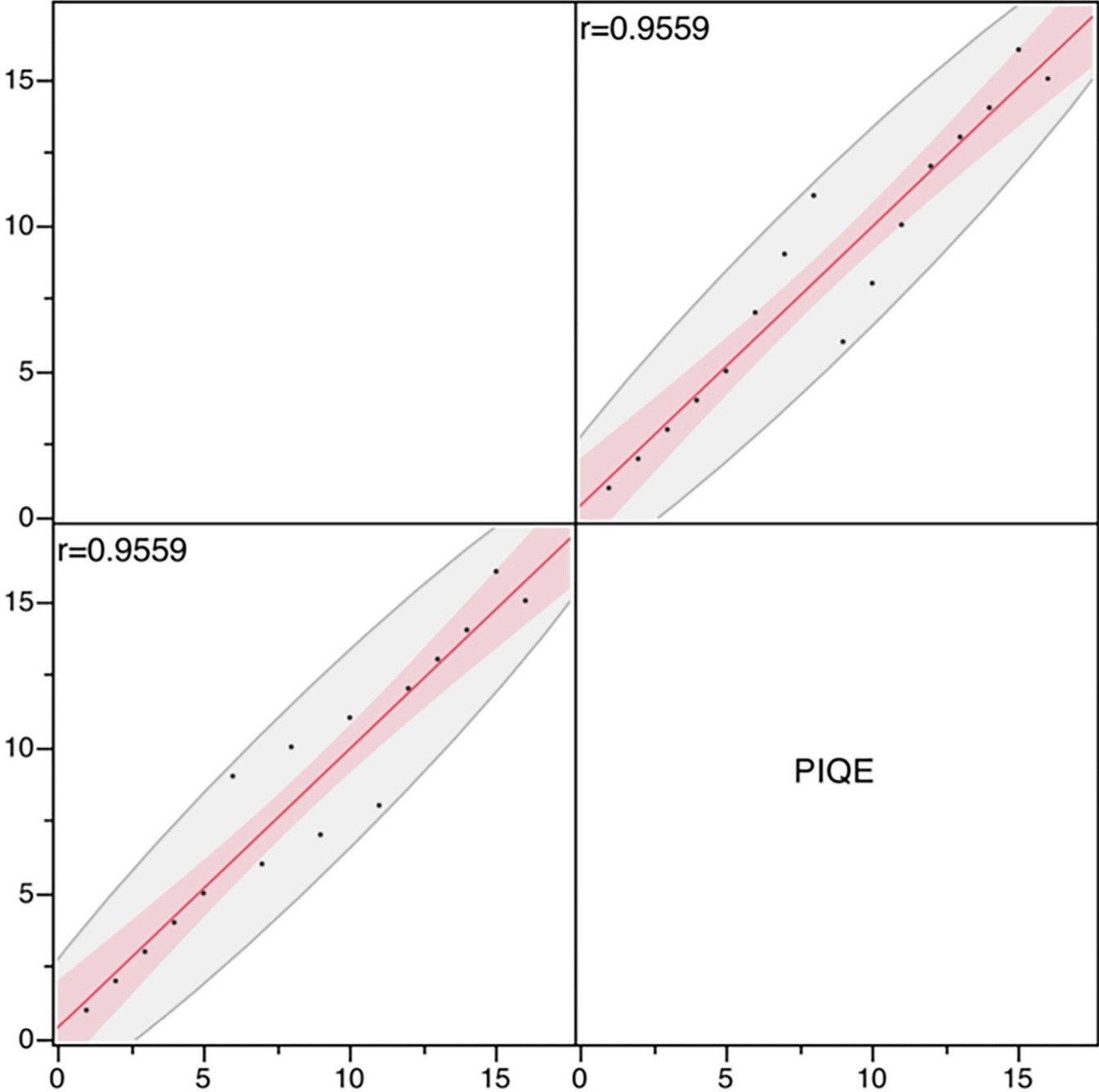

**Fig 7. Correlation between the visual assessment and PIQE rankings.** PIQE: Perception-based image quality evaluator. Spearman's significant difference test between the visual assessment and PIQE rankings revealed a rs of 0.9559 (p < 0.0001), indicating a strong correlation.

was compared with the NMSE, and a statistical correlation was determined. Evaluation using PIQE demonstrated a strong correlation with visual assessment, suggesting equivalence between these two methods.

The results ranked by evaluators 6 and 7 were inconsistent compared to the other evaluators. Consequently, their evaluations were excluded, underscoring the validity of the

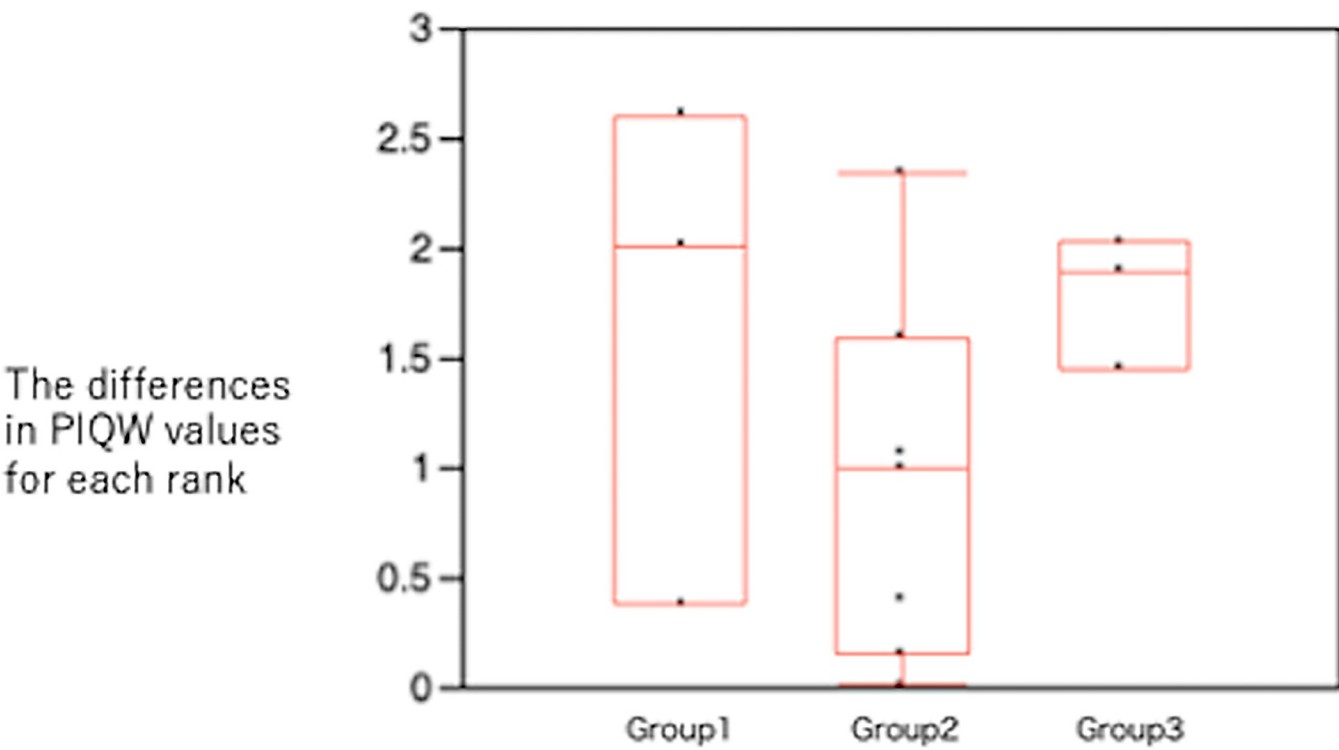

**Fig 8. Significant difference examined in the ranking of PIQE.** The difference between the top and bottom ranks from 1st to 16th. The difference between 1st and 4th place was group 1, the difference between 5th and 12th place was group 2, and the difference between 13th and 16th place was group 3. There was no significant difference among the three groups.

**Table 8. NIQE scores and their ranking.**

| Target image | NIQE score | Rank |
| --- | --- | --- |
| 880-1800s | 6.6500 | 2 |
| 440-1800s | 7.3372 | 11 |
| 880-900s | 6.9129 | 8 |
| 440-900s | 6.7181 | 4 |
| 880-600s | 7.1605 | 10 |
| 440-600s | 6.7528 | 5 |
| 880-450s | 7.3985 | 12 |
| 440-450s | 6.9369 | 9 |
| 880-360s | 7.6248 | 13 |
| 440-360s | 6.6778 | 3 |
| 880-300s | 7.8463 | 14 |
| 440-300s | 6.8895 | 7 |
| 880-180s | 8.0195 | 15 |
| 440-180s | 6.8434 | 6 |
| 880-120s | 8.1071 | 16 |
| 440-120s | 6.6168 | 1 |

NIQE: Natural Image Quality Evaluator; Lower scores represent better image quality.

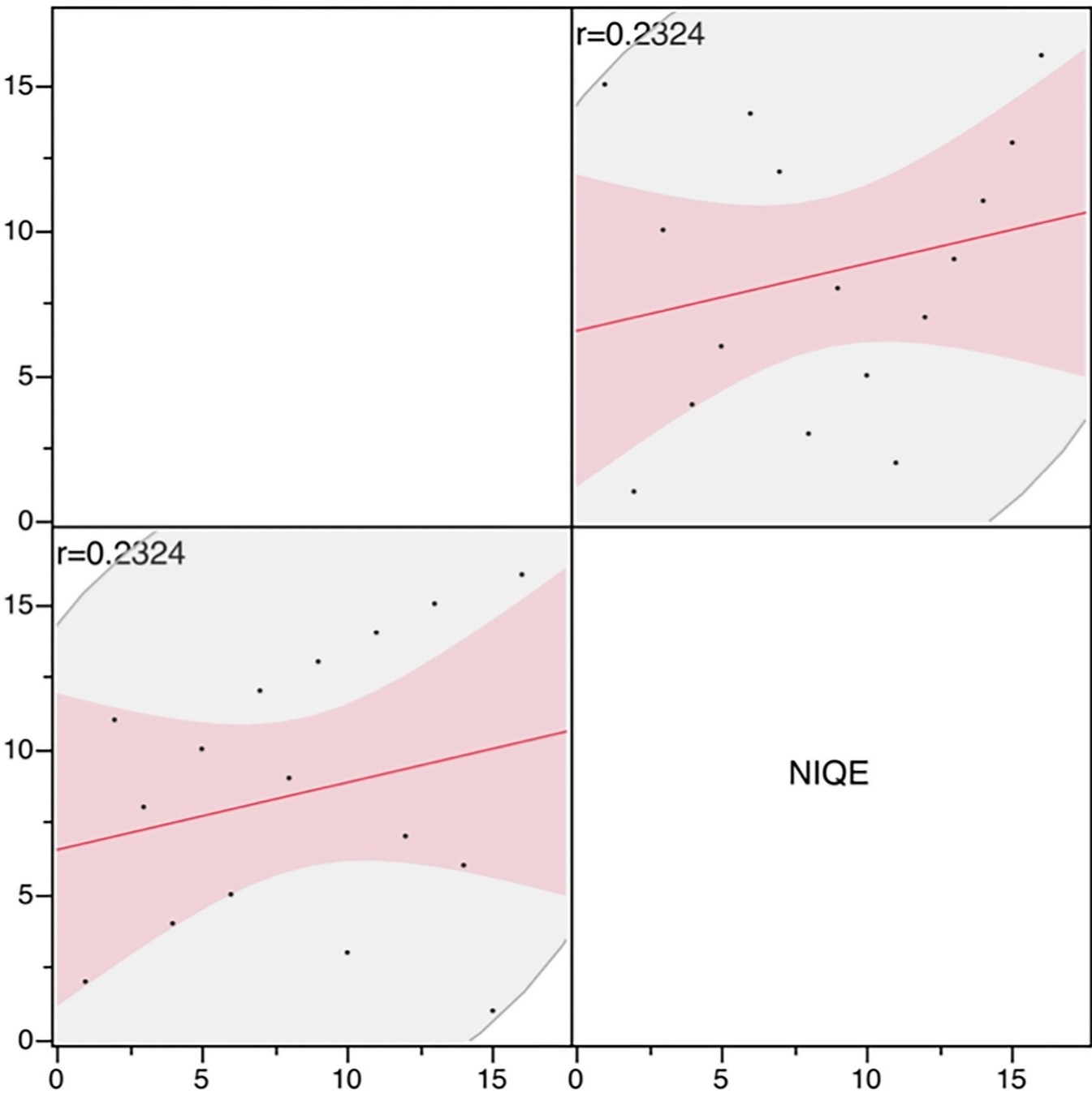

**Fig 9. Correlation between the visual assessment and NIQE rankings.** NIQE: natural image quality evaluator. Spearman's significant difference test between the visual assessment and NIQE rankings revealed a rs of 0.2324 (p 0.3865), indicating no strong correlation between the two methods.

evaluators' selection. This also underscores that the evaluation criteria are not easily applied by any evaluator.

Because NMSE evaluates the target image using a reference image, it is generally impossible to evaluate images with different numbers of pixels. In this study, images with different acquisition times were evaluated using NMSE scores, using different references for 880- and 440-pixel images.

**Table 9. Results of the uniformity evaluation medulla of the frontal lobe.**

| CV (440) | | CV (880) | |
|---|---|---|---|
| 440-60s | 8.02 | 880-60s | 8.48 |
| 440-120s | 8.67 | 880-120s | 9.20 |
| 440-180s | 8.22 | 880-180s | 8.49 |
| 440-300s | 8.03 | 880-300s | 7.52 |
| 440-360s | 7.75 | 880-360s | 7.2 |
| 440-450s | 5.87 | 880-450s | 5.24 |
| 440-600s | 4.70 | 880-600s | 4.19 |
| 440-900s | 4.27 | 880-900s | 3.79 |
| 440-1800s | 2.98 | 880-1800s | 2.65 |

CV: coefficient of variation; Uniformity was evaluated numerically using the ROI (5 mm in diameter) placed on the medulla of the frontal lobe.

From the PIQE results, if the proportion of statistical noise was approximately the same, higher resolution was associated with higher evaluation. This trend was also reflected in visual assessments, indicating the potential for objective evaluation of not only statistical noise but

**Table 10. Results of the uniformity evaluation medulla of the temporal lobe.**

| CV (440) | | CV (880) | |
|---|---|---|---|
| 440-60s | 20.95 | 880-60s | 20.74 |
| 440-120s | 9.73 | 880-120s | 9.41 |
| 440-180s | 7.04 | 880-180s | 7.13 |
| 440-300s | 6.37 | 880-300s | 6.6 |
| 440-360s | 4.89 | 880-360s | 5.14 |
| 440-450s | 5.40 | 880-450s | 5.67 |
| 440-600s | 4.79 | 880-600s | 5.06 |
| 440-900s | 4.10 | 880-900s | 4.03 |
| 440-1800s | 3.33 | 880-1800s | 3.07 |

CV: coefficient of variation; Uniformity was evaluated numerically using the ROI (5 mm in diameter) placed on the medulla of the temporal lobe.

**Table 11. Results of the uniformity evaluation medulla of the occipital lobe.**

| CV (440) | | CV (880) | |
|---|---|---|---|
| 440-60s | 23.71 | 880-60s | 24.31 |
| 440-120s | 9.63 | 880-120s | 9.64 |
| 440-180s | 7.52 | 880-180s | 7.43 |
| 440-300s | 8.01 | 880-300s | 8.07 |
| 440-360s | 7.88 | 880-360s | 7.80 |
| 440-450s | 7.09 | 880-450s | 7.12 |
| 440-600s | 6.41 | 880-600s | 6.54 |
| 440-900s | 5.14 | 880-900s | 5.24 |
| 440-1800s | 2.73 | 880-1800s | 2.8 |

CV: coefficient of variation; Uniformity was evaluated numerically using the ROI (5 mm in diameter) placed on the medulla of the occipital lobe.

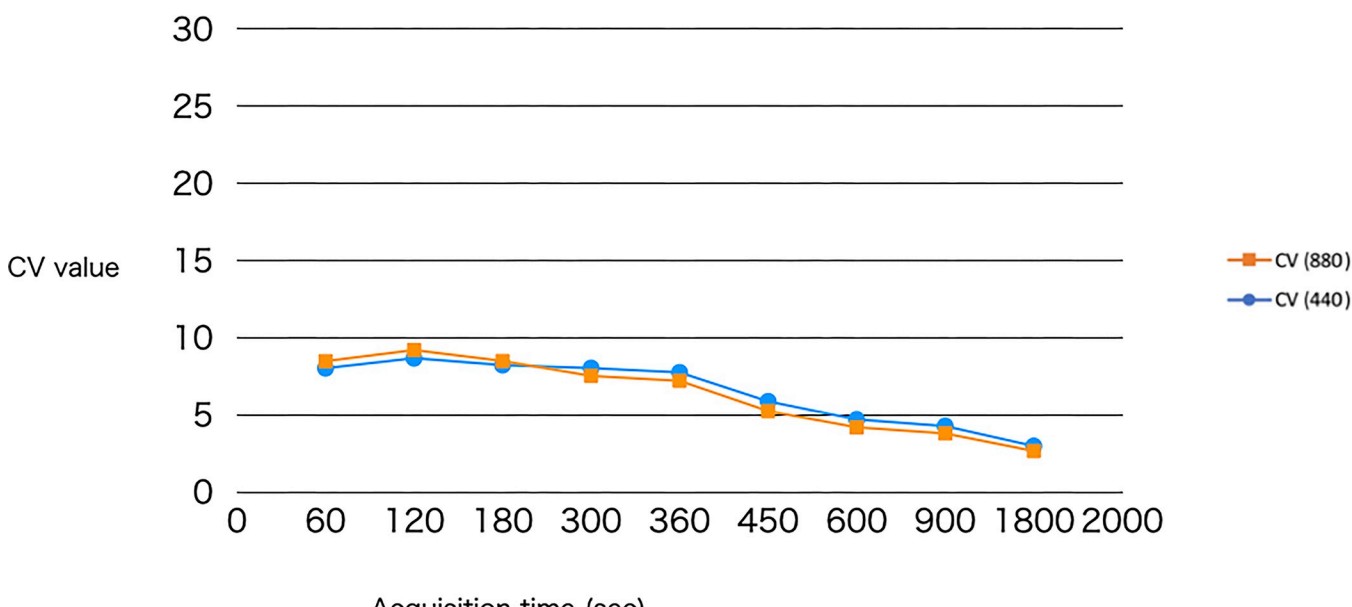

front

**Fig 10. Graphical display of the results of Table 9.**

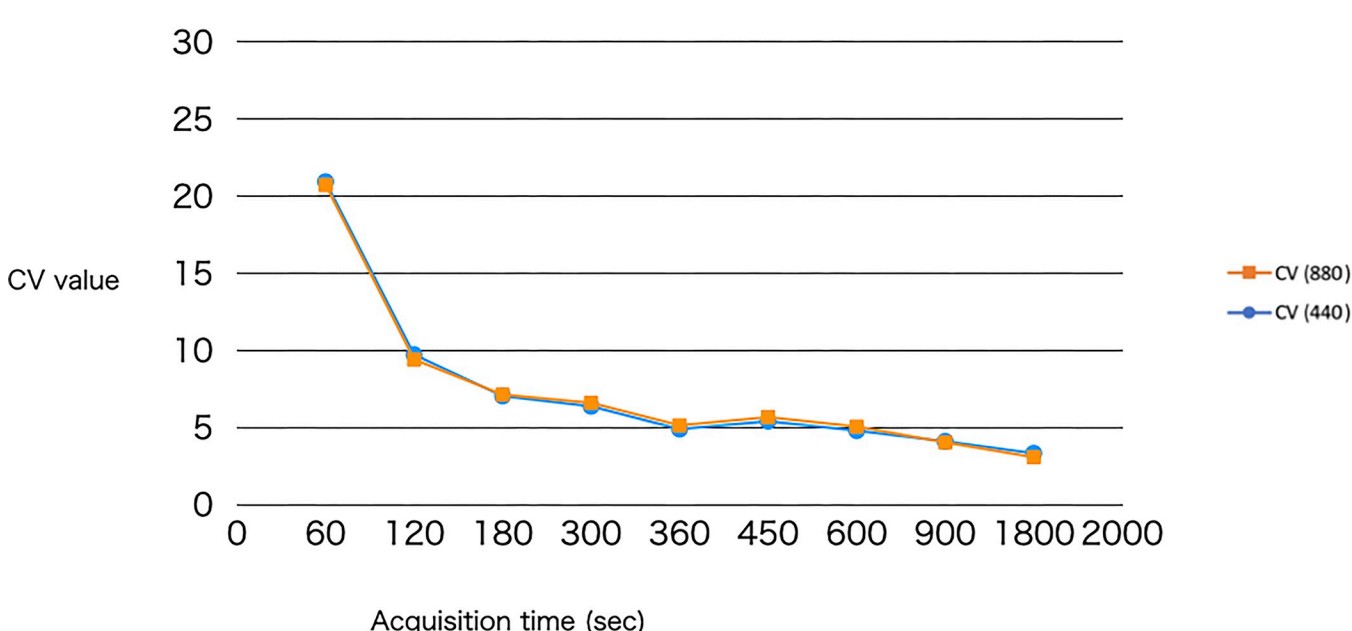

tempo

**Fig 11. Graphical display of the results of Table 10.**

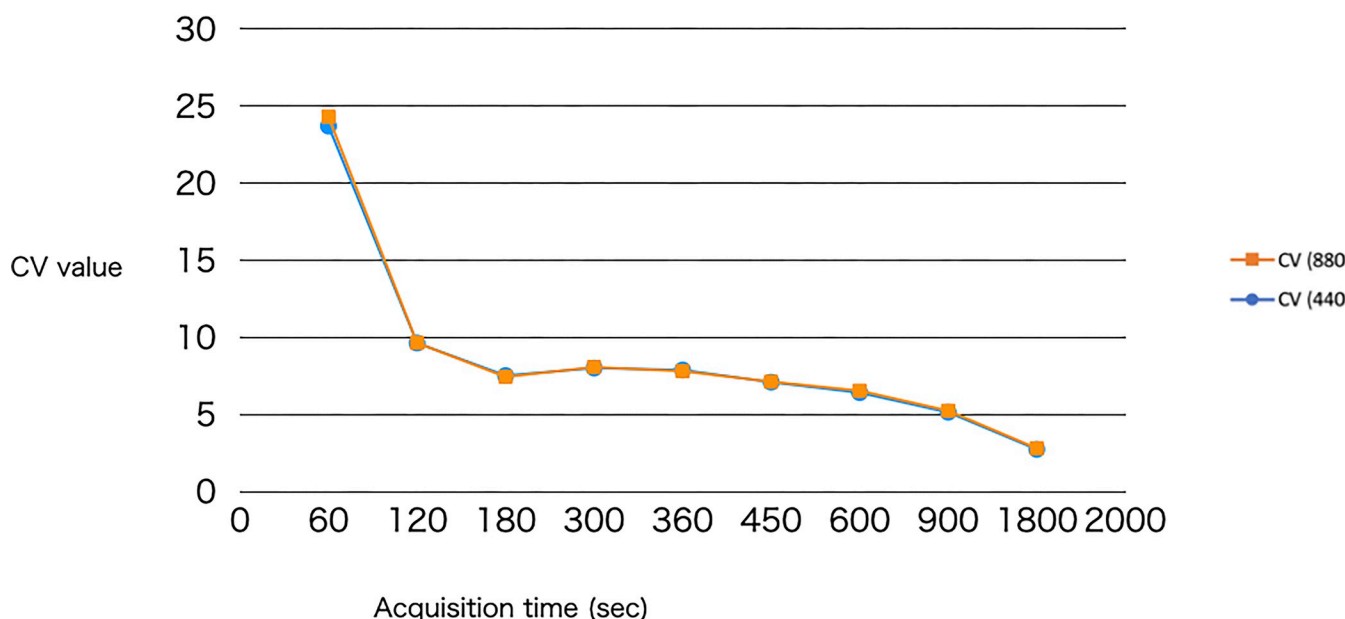

**Fig 12. Graphical display of the results of Table 11.**

also resolution differences using PIQE. The images ranked 1st to 4th in the PIQE results in Table 7 are arranged in the correct order reflecting the acquisition time and the number of pixels. It also agrees with the results of visual evaluation in Table 1. The 13th to 16th low-quality images in Table 7 also reflect the acquisition time and the number of pixels. Although there is a ranking reversal between the 15th and 16th images in visual assessment, it pertains to a low-ranking image, typically not accepted in clinical imaging. The discrepancy in the visual evaluation ranking by the image experts is believed to be due to the unfamiliarity with low-resolution images. In the ranking of 5th to 12th from 880-600s to 440-300s, PIQE provided better results for images with a higher number of pixels compared to the acquisition time. Within this range, the images were arranged in order of acquisition time. Visual evaluation by an image expert revealed that the order of pixel count and acquisition time matched, unlike the results obtained from PIQE. It is considered that sharpness is prioritized over noise in this image quality range. Although there was a difference between the results of the visual evaluation and the PIQE ranking within this range, there was no significant difference in the ranking in the Spearman's significance test, and it is considered that the PIQE has the same evaluation ability as the visual evaluation. In addition, we calculated the difference between the bottom and top rankings in PIQE values, that is, 880-1800s to 440-900s ranked 1st to 4th, 880-600s to 440-300s ranked 5th to 12th, and 880s-180s to 440-120s ranked 13th to 15th. Intergroup comparison was performed by Mann-Whitney's U test in three groups, there was no significant difference among them. It is thought that PIQE demonstrated the capability to evaluate the image quality of the 880-600s to 440-300s, ranked 5th to 12th, at a level comparable to that of visual evaluation.

Uniformity was evaluated, and as shown in Figs 8, 9 and 10, both 440- and 880-pixel images proved that the longer the imaging time, the higher the uniformity of the image. The results were almost consistent with the visual evaluation.

Visual evaluation of images from 1800 to 180 s showed that a longer acquisition time resulted in better evaluation scores (Table 3). The results and rankings obtained using NMSE

were similar (Table 6). For images with the same acquisition time, 880-pixel images scored better than 440-pixel images (Table 3).

For images with an acquisition time of 120 s, the difference in ranking between 440 and 880 pixels was less than 0.2 points, which is a much smaller difference compared with that of the other rankings; however, it reversed the visual evaluation rankings (Table 3).

Evaluators 4 and 5 evaluated the ranking of 440- and 880-pixel images with a 120 s acquisition time, reversing the rating order for items 1 and 2 (Tables 1 and 2). They were diagnostic radiologists with more than 10 years of clinical experience and nuclear medicine specialists. This evaluation reversed the average rankings for the 440- and 880-pixel images at 120 s. For item 1, both evaluators found that the boundary between the white matter and gray matter of the temporal lobe and the peripapillary thalamus was clearer in the 440-pixel image because it had a wider area without accumulation. For item 2, the 440-pixel image showed more uniform accumulation because of a denser accumulation in the frontotemporal white matter, thalamus, and caudate nucleus in general. Noisy images acquired at 120 s were not of optimal quality for use in clinical imaging.

The rankings obtained from visual evaluation and the no-reference metric method were compared by five evaluators. Generally, supervised methods outperform unsupervised methods [26]. However, when creating a dataset for supervised learning in nuclear medicine, which is not well standardized, generating a standard image is not easy [13, 14]. It is more realistic to perform a general-purpose quantitative evaluation using supervised learning rather than a target [13, 14]. In this study, PIQE, an unsupervised method that does not require training data to evaluate image quality, yielded better results [15, 27]. Moreover, since PIQE does not depend on training data, it is considered a less environmentally dependent metric that can be handled on the same scale at all facilities conducting nuclear medicine examinations and imaging. Hence, PIQE may be an efficient image evaluation method.

The NIQE results showed no correlation with the visual evaluation results. This could be because NIQE is a supervised method that employs a learning model using natural scene statistics [17].

Similar to natural images, PET images follow the Poisson distribution for image generation [32]. Because no-reference quality metrics match subjective human quality scores over fully referenced quality metrics, PET image evaluation using a no-reference metric was expected to be useful. This is another reason why PIQE is more consistent with visual evaluation than NIQE.

Numerical evaluation plays an important role in the image evaluation and medical treatment fields [33, 34]. Studies have conducted various evaluations without setting a gold standard. While various evaluation methods exist without setting a gold standard, methods without reference images are expected to gain wider acceptance for scoring and ranking image quality in the future [35, 36].

Despite its strengths, this study had limitations, notably the absence of clinical imaging based on brain phantom images. Nonetheless, our findings suggest that PIQE may be comparable to visual evaluation by radiologists and specialists, offering potential applications in clinical image evaluation across various anatomical regions.

## Conclusions

In conclusion, this study investigated the application of no-reference metrics, specifically PIQE, in evaluating image quality for clinical images in nuclear medicine. The results demonstrate that PIQE evaluations align closely with visual evaluations by specialists, suggesting its potential as a reliable method for clinical image quality assessment. Moving forward,

additional research and validation are warranted to fully integrate no-reference metrics into routine clinical practice in nuclear medicine.

## Acknowledgments

We are grateful to the radiological technicians at the Department of Radiology, Osaka Metropolitan University Hospital.

## Author Contributions

**Conceptualization:** Shigeaki Higashiyama, Yutaka Katayama, Atsushi Yoshida.

**Data curation:** Shigeaki Higashiyama, Yutaka Katayama, Atsushi Yoshida, Nahoko Inoue, Takashi Yamanaga, Takao Ichida, Joji Kawabe.

**Formal analysis:** Shigeaki Higashiyama, Yukio Miki, Joji Kawabe.

**Project administration:** Yukio Miki.

**Supervision:** Takao Ichida, Yukio Miki.

**Writing – original draft:** Shigeaki Higashiyama.

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
