## [Decision Letter · Decision Letter 0]

15 Mar 2024

PONE-D-23-28461Investigation of the effectiveness of No-reference Metric in Image evaluation in Nuclear MedicinePLOS ONE

Dear Dr. Higashiyama,

Thank you for submitting your manuscript to PLOS ONE. After careful consideration, we feel that it has merit but does not fully meet PLOS ONE’s publication criteria as it currently stands. Therefore, we invite you to submit a revised version of the manuscript that addresses the points raised during the review process.

We look forward to receiving your revised manuscript.

Kind regards,

Sadiq H. Abdulhussain, Ph.D.

Academic Editor

PLOS ONE

Journal Requirements:

4. PLOS requires an ORCID iD for the corresponding author in Editorial Manager on papers submitted after December 6th, 2016. Please ensure that you have an ORCID iD and that it is validated in Editorial Manager. To do this, go to ‘Update my Information’ (in the upper left-hand corner of the main menu), and click on the Fetch/Validate link next to the ORCID field. This will take you to the ORCID site and allow you to create a new iD or authenticate a pre-existing iD in Editorial Manager. Please see the following video for instructions on linking an ORCID iD to your Editorial Manager account: https://www.youtube.com/watch?v=_xcclfuvtxQ".

5. We note that your Data Availability Statement is currently as follows: [All relevant data are within the manuscript and its Supporting Information files.]

6. We note that Figure 1 and 4 in your submission contain copyrighted images. All PLOS content is published under the Creative Commons Attribution License (CC BY 4.0), which means that the manuscript, images, and Supporting Information files will be freely available online, and any third party is permitted to access, download, copy, distribute, and use these materials in any way, even commercially, with proper attribution. For more information, see our copyright guidelines: http://journals.plos.org/plosone/s/licenses-and-copyright.

1. You may seek permission from the original copyright holder of Figure 1 and 4 to publish the content specifically under the CC BY 4.0 license.

Reviewers' comments:

Reviewer's Responses to Questions

**Comments to the Author**

1. Is the manuscript technically sound, and do the data support the conclusions?

Reviewer #1: Partly

Reviewer #2: Yes

2. Has the statistical analysis been performed appropriately and rigorously? 

Reviewer #1: N/A

Reviewer #2: Yes

3. Have the authors made all data underlying the findings in their manuscript fully available?

Reviewer #1: Yes

Reviewer #2: Yes

4. Is the manuscript presented in an intelligible fashion and written in standard English?

Reviewer #1: No

Reviewer #2: No

5. Review Comments to the Author

Reviewer #1: (No Response)

Reviewer #2: This paper examines whether No-Reference metrics can be applied to image quality evaluations for clinical images in nuclear medicine. To this end, 14 images with different numbers of pixels and acquisition times were created and extracted from Biograph Vision. The paper is well organized but not well written. I have some minor points that need to be addressed.

1. The contribution is not clear and needs to be highlighted.

2. The introduction should be modified so that the last paragraphs should be divided as paper contributions and finds and paper organization as a separated subsections.

3. Please make the contributions as a bullet points.

4. The novelty of the paper needs to be modified to be more clear.

5. Some grammar errors are discovered so please try to proofread the paper again.

6. I would suggest make it clear what have been done in comparison to the state of the art methods showing the contribution of the paper.

7. I would suggest adding equation number.

8. I have found that there is an evaluation before the results section. I would suggest making the evaluation in results section part to make it clear to the reader.

9. The author state that (The total score in shown in bottom row of..) what dose that means please correct.

10. Tables should not be splatted into two pages please correct table 14.

11. There are many references in the discussion part after the results. Why not stated these references in the introduction part.

12. The conclusion is too short and not clear please modify.

6. PLOS authors have the option to publish the peer review history of their article (what does this mean?). If published, this will include your full peer review and any attached files.

Reviewer #1: None

Reviewer #2: No

---

## [Author Response · Author response to Decision Letter 0]

4 Apr 2024

Thank you for your careful review and your comments.

We have made changes to comply with your suggestions. The revised parts of the manuscript are written in red.

Journal Requirements:

1 When submitting your revision, we need you to address these additional requirements.

→We have made the changes according to your suggestions.

2 Note from Emily Chenette, Editor in Chief of PLOS ONE, and Iain Hrynaszkiewicz, Director of Open Research Solutions at PLOS: Did you know that depositing data in a repository is associated with up to a 25% citation advantage (https://doi.org/10.1371/journal.pone.0230416)? If you’ve not already done so, consider depositing your raw data in a repository to ensure your work is read, appreciated and cited by the largest possible audience. 

→ We have read and understood the contents.

3 Please note that PLOS ONE has specific guidelines on code sharing for submissions in which author-generated code underpins the findings in the manuscript. In these cases, all author-generated code must be made available without restrictions upon publication of the work. 

→ We have read and understood the contents.

4 PLOS requires an ORCID iD for the corresponding author in Editorial Manager on papers submitted after December 6th, 2016.

→ The authors and corresponding author (Shigeaki Higashiyama) are registered ORCID members.

5 We note that your Data Availability Statement is currently as follows: [All relevant data are within the manuscript and its Supporting Information files.]

→ All relevant data are present in the manuscript and Supporting Information files.

6 We note that Figure 1 and 4 in your submission contain copyrighted images. All PLOS content is published under the Creative Commons Attribution License (CC BY 4.0), which means that the manuscript, images, and Supporting Information files will be freely available online, and any third party is permitted to access, download, copy, distribute, and use these materials in any way, even commercially, with proper attribution. For more information, see our copyright guidelines: http://journals.plos.org/plosone/s/licenses-and-copyright.　We require you to either (1) present written permission from the copyright holder to publish these figures specifically under the CC BY 4.0 license, or (2) remove the figures from your submission:

→ Figures 1 and 4 are images that I created and used in this paper. This is not a quotation from any literature or book. I had inquired about permission to use the preprint without peer review, but I received a reply stating that there was no need to confirm the license. I have added that email as a PDF attachment to the other file.

Comments to the Author　

1-3 and 6 → We do not require a response.

4→ We utilized a paid English proofreading of the main text. Certificate of editing was attached the file as PDF.

5 Review Comments to the Author

To Reviewer #1: (No Response)

To Reviewer #2 

1、 The contribution is not clear and needs to be highlighted.

→ We have modified the introduction to include a subsection specifically dedicated to outlining the contributions and findings of the paper.

2、The introduction should be modified so that the last paragraphs should be divided as paper contributions and finds and paper organization as a separated subsections.

→ We divided the Introduction into paragraphs on Contributions and findings.

3 Please make the contributions as a bullet points.

→ We divided the Introduction and Abstract into paragraphs and clearly stated the novelty. 

4、The novelty of the paper needs to be modified to be more clear.

→ A sentence explaining the novelty of this research has been added to the introduction.

5、Some grammar errors are discovered so please try to proofread the paper again.

→ We have used a paid English proofreading service. Certificate of editing was attached the file as PDF.

6、 I would suggest make it clear what have been done in comparison to the state of the art methods showing the contribution of the paper.

→ In the introduction, we have added a description on the novel aspects of this study compared to what has been presented in past papers.

7、 I would suggest adding equation number.

→ We have numbered the equations in the text.

8、 I have found that there is an evaluation before the results section. I would suggest making the evaluation in results section part to make it clear to the reader.

→ Based on you suggestion, we have made changes to make the results easier to understand.

9、The author state that (The total score in shown in bottom row of..) what dose that means please correct.

→ We have added a description of the table you pointed out. 

10、 Tables should not be splatted into two pages please correct table 14.

→ We have accordingly reorganized Table 14.

11、There are many references in the discussion part after the results. Why not stated these references in the introduction part.

→ Some of the references for the discussion are cited in the methods section.

12、The conclusion is too short and not clear please modify.

→As pointed out, I have further elaborated the content in the conclusion section.

---

## [Decision Letter · Decision Letter 1]

24 Jun 2024

PONE-D-23-28461R1Investigation of the Effectiveness of No-reference Metric in Image Evaluation in Nuclear MedicinePLOS ONE

Dear Dr. Higashiyama,

Thank you for submitting your manuscript to PLOS ONE. After careful consideration, we feel that it has merit but does not fully meet PLOS ONE’s publication criteria as it currently stands. Therefore, we invite you to submit a revised version of the manuscript that addresses the points raised during the review process.

We look forward to receiving your revised manuscript.

Kind regards,

Sadiq H. Abdulhussain, Ph.D.

Academic Editor

PLOS ONE

Additional Editor Comments:

The authors are asked to check the attached files for reviewer comments.

Reviewers' comments:

Reviewer's Responses to Questions

**Comments to the Author**

1. If the authors have adequately addressed your comments raised in a previous round of review and you feel that this manuscript is now acceptable for publication, you may indicate that here to bypass the “Comments to the Author” section, enter your conflict of interest statement in the “Confidential to Editor” section, and submit your "Accept" recommendation.

Reviewer #1: (No Response)

Reviewer #2: All comments have been addressed

2. Is the manuscript technically sound, and do the data support the conclusions?

Reviewer #1: Partly

Reviewer #2: Yes

3. Has the statistical analysis been performed appropriately and rigorously? 

Reviewer #1: I Don't Know

Reviewer #2: Yes

4. Have the authors made all data underlying the findings in their manuscript fully available?

Reviewer #1: Yes

Reviewer #2: Yes

5. Is the manuscript presented in an intelligible fashion and written in standard English?

Reviewer #1: No

Reviewer #2: Yes

6. Review Comments to the Author

Reviewer #1: (No Response)

Reviewer #2: All the comments have been addressed. The authors modified the paper based on the suggested comment. Iwouls suggest accept this paper.

7. PLOS authors have the option to publish the peer review history of their article (what does this mean?). If published, this will include your full peer review and any attached files.

Reviewer #1: No

Reviewer #2: No

---

## [Author Response · Author response to Decision Letter 1]

26 Jun 2024

Response to Reviewers

1. If the authors have adequately addressed your comments raised in a previous round of review and you feel that this manuscript is now acceptable for publication, you may indicate that here to bypass the “Comments to the Author” section, enter your conflict of interest statement in the “Confidential to Editor” section, and submit your "Accept" recommendation.

Reviewer #1: (No Response) → We have corrected the points raised in the point by point response.

Reviewer #2: All comments have been addressed → Thank you for reviewing the revised manuscript. 

2. Is the manuscript technically sound, and do the data support the conclusions?

Reviewer #1: Partly　→ Thank you for reviewing the revised manuscript. We have corrected the points raised in the point by point response. 

Our study uses phantom data that is accessible from any research institution, not patient clinical data. Therefore, reproducing the methods described in the manuscript is feasible and technically sound. Consequently, we believe that faithfully replicating the methods and evaluation criteria of this paper allows for accurate replication of the conclusions presented in the manuscript. As such, no changes or additions to the evaluation methods or data have been made.

Reviewer #2: Yes　→ Thank you for reviewing the revised manuscript. 

3. Has the statistical analysis been performed appropriately and rigorously? 

Reviewer #1: I Don't Know　→ We have corrected the points raised in the point by point response.　Our study employed JMP (SAS Japan Co., Ltd) for statistical analysis. Therefore, the statistical analysis in the manuscript is replicable, and technically robust. 

Reviewer #2: Yes　→ Thank you for reviewing the revised manuscript. 

4. Have the authors made all data underlying the findings in their manuscript fully available?

Reviewer #1: Yes　→Thank you for reviewing the revised manuscript. 

Reviewer #2: Yes　→Thank you for reviewing the revised manuscript. 

5. Is the manuscript presented in an intelligible fashion and written in standard English?

Reviewer #1: No　→We have corrected the points raised in the point by point response.　 In addition, we have followed the points raised in the first peer review.　We have engaged a professional medical manuscript editing service to revise the manuscript at our own expense. The revised manuscript, addressing these comments, is now submitted. If there are any parts that do not conform to standard English, please provide specific feedback, and we will promptly make corrections. 

Certificate of English ProofreadingHonyaku Center Inc. certifies that the manuscript entitled

Investigation of the Effectiveness of No-reference Metric in Image Evaluation in Nuclear Medicine

has been edited and corrected to the highest standards.

Neither the contents of this manuscript nor the author’s intentions have been altered in any way.

This manuscript has been edited and corrected by an experienced proofreader who is a native speaker of

English and who is under the direct supervision of Honyaku Center Inc.

ISSUED ON　March 28, 2024　AUTHORS　Shigeaki Higashiyama　JOB CODE　BQYRQ_2　Yasuo Terashima

Reviewer #2: Yes　→Thank you for reviewing the revised manuscript. We will proceed with the submission as it is.

6. Review Comments to the Author

Reviewer #1: (No Response)　 →Thank you for reviewing the revised manuscript. We will proceed with the submission as it is.

Reviewer #2: All the comments have been addressed. The authors modified the paper based on the suggested comment. Iwouls suggest accept this paper.　 →Thank you for reviewing the revised manuscript. We will proceed with the submission as it is.

Point-by-point response

・98: 2 6 MBq basis for inclusion.

⇒The Japanese Society of Nuclear Medicine has publicly released a procedure manual for phantom tests. We used the radiation dose from the FDG and Amyloid Brain PET Imaging Phantom Test Procedure Manual (in Japanese). https://jsnm.org/wp_jsnm/wp-content/uploads/2021/02/Dementia_PhantomTest_20210208.pdf

・99：PET/CT scannerのBiograph visionは450？600？

・99: The PET/CT scanner used was the Biographic vision 450 or 600?

⇒ We have made changes to lines 86 and 97.

・112：Pixel size 0.825 mm is only correct for a matrix size of 880 when the transverse FOV is 726 mm.

⇒ With an FOV of 363 mm, for a Matrix Size of 440, the voxel size becomes 0.825 mm.

We have added a description of the FOV in the 99th line.

・113: Is 3 mm correct for Slice thickness instead of 3 cm?

⇒ We have made changes to lines 101.

116: Imaging conditions for computed tomography attenuation correction should be stated.

　　The description of Random correction should also be added.

⇒ We have described the reconstruction conditions using CT and the random correction in lines 104 and 111.

・120: 440 matrix size is more appropriate than 440 pixel size.

⇒ We have made changes to lines 109,110.

 ：What does 0.4125 mm×0.4125×2 mm３mean? If the transverse FOV is 726 mm and the matrix size is 440, the pixel size is 1.65 mm; if the matrix size is 880, the pixel size is 0.825 mm. If the magnification factor is 2x, the pixel size is 0.825 mm for a 440 matrix size and 0.4125 for an 880 matrix size.

⇒ The FOV is 363 mm. We have added this to line 100.

・122: Statistical analysis instruments are described, but significance levels are not stated.　 

⇒ We have added p-value settings to lines 217 and 220.

・235：CV should be assessed on a uniform phantom. There is also poor evidence for the size and location of the ROI; if CV is measured once in each series, it should be measured at least three times to account for measurement error.

⇒ We evaluate with ROIs of 5 mm diameter when conducting clinical image evaluations at our facility. In line with that evaluation method, we conducted evaluations using ROIs of 5 mm diameter. This information has been added to line 226.

⇒ We did not conduct experiments with a pool phantom because we aimed for evaluations based on images intended for clinical examinations.

・394: acquisition time is appropriate, not capturing time.

⇒ We have made changes to lines 381.

・The matrix size and color scale should be added to Fig 1 and Fig 2 respectively.

⇒We have modified lines 566 to 572.

・The names of the vertical axes in Fig 6 should be added.

⇒ We have made corrections to lines 218 to 221 and added them to Fig. 6.

・The names of the vertical and horizontal axes in Figure 8 are mandatory. If the vertical axis is the CV value and the horizontal axis is the acquisition time, they should be described.

⇒We have made modifications to Figure 8.

・Is the value obtained for the 440 and 880 matrix size for 330 s in the CV results in Table 4 correct?

⇒Thank you very much for your detailed feedback. There are no mistakes in the parts you pointed out, and the results are accurate.

・The CV values were similar for the 880 matrix size and the 440 matrix size. However, the visual evaluation results show that the 880 matrix size is better for all acquisition time when compared to the 440 matrix size.

⇒Thank you very much for your detailed feedback. There are no mistakes in the parts you pointed out, and the results are accurate.

---

## [Decision Letter · Decision Letter 2]

6 Aug 2024

PONE-D-23-28461R2Investigation of the Effectiveness of No-reference Metric in Image Evaluation in Nuclear MedicinePLOS ONE

Dear Dr. Higashiyama,

Thank you for submitting your manuscript to PLOS ONE. After careful consideration, we feel that it has merit but does not fully meet PLOS ONE’s publication criteria as it currently stands. Therefore, we invite you to submit a revised version of the manuscript that addresses the points raised during the review process.

We look forward to receiving your revised manuscript.

Kind regards,

Sadiq H. Abdulhussain, Ph.D.

Academic Editor

PLOS ONE

Journal Requirements:

Reviewers' comments:

Reviewer's Responses to Questions

**Comments to the Author**

1. If the authors have adequately addressed your comments raised in a previous round of review and you feel that this manuscript is now acceptable for publication, you may indicate that here to bypass the “Comments to the Author” section, enter your conflict of interest statement in the “Confidential to Editor” section, and submit your "Accept" recommendation.

Reviewer #1: All comments have been addressed

Reviewer #3: All comments have been addressed

2. Is the manuscript technically sound, and do the data support the conclusions?

Reviewer #1: Partly

Reviewer #3: Yes

3. Has the statistical analysis been performed appropriately and rigorously? 

Reviewer #1: I Don't Know

Reviewer #3: Yes

4. Have the authors made all data underlying the findings in their manuscript fully available?

Reviewer #1: Yes

Reviewer #3: Yes

5. Is the manuscript presented in an intelligible fashion and written in standard English?

Reviewer #1: Yes

Reviewer #3: Yes

6. Review Comments to the Author

Reviewer #1: (No Response)

Reviewer #3: (No Response)

7. PLOS authors have the option to publish the peer review history of their article (what does this mean?). If published, this will include your full peer review and any attached files.

Reviewer #1: No

Reviewer #3: No

---

## [Author Response · Author response to Decision Letter 2]

7 Aug 2024

Thank you for your careful review and your comments.

We have made changes to comply with your suggestions.

Changed parts are written in red.

Reviewer Comments

84：Additional references to the guidelines.

→References have been added and additional information has been added.

100, 112, 113, 115：Half-width space added between number and unit.

→We made the changes as you suggested.

125: 880 matrix size is more correct than 880 pixels. Please check the other text as well; Fig. 1a should be a PET image with 440 matrix size, not 440 s.

→We made the changes as you suggested.

We evaluate with ROIs of 5 mm diameter when conducting clinical image evaluations

at our facility.

⇒ In the diagnostic imaging, it is necessary to state the validity of the 'method of placing a 5 mm ROI on the brain PET image', as shown in Fig. 4. For example, the rationale based on guidelines and previous studies. 

→References have been added and additional information has been added.

The number of measurements should be performed at least three times due to the variability of the data.

→　To check the variability of the data, we examined it in three locations: the frontal lobe, the temporal lobe, and the occipital lobe. Changes and additions to the table and figure have been added at lines 226、331 and 387.

Addition of units for acquisition time in Fig. 8.

→We made the changes as you suggested.

---

## [Decision Letter · Decision Letter 3]

29 Aug 2024

Investigation of the Effectiveness of No-reference Metric in Image Evaluation in Nuclear Medicine

PONE-D-23-28461R3

Dear Dr. Higashiyama,

We’re pleased to inform you that your manuscript has been judged scientifically suitable for publication and will be formally accepted for publication once it meets all outstanding technical requirements.

Kind regards,

Sadiq H. Abdulhussain, Ph.D.

Academic Editor

PLOS ONE

Additional Editor Comments (optional):

Reviewers' comments:

Reviewer's Responses to Questions

**Comments to the Author**

1. If the authors have adequately addressed your comments raised in a previous round of review and you feel that this manuscript is now acceptable for publication, you may indicate that here to bypass the “Comments to the Author” section, enter your conflict of interest statement in the “Confidential to Editor” section, and submit your "Accept" recommendation.

Reviewer #1: All comments have been addressed

Reviewer #3: All comments have been addressed

2. Is the manuscript technically sound, and do the data support the conclusions?

Reviewer #1: Yes

Reviewer #3: Yes

3. Has the statistical analysis been performed appropriately and rigorously? 

Reviewer #1: Yes

Reviewer #3: Yes

4. Have the authors made all data underlying the findings in their manuscript fully available?

Reviewer #1: Yes

Reviewer #3: Yes

5. Is the manuscript presented in an intelligible fashion and written in standard English?

Reviewer #1: Yes

Reviewer #3: Yes

6. Review Comments to the Author

Reviewer #1: (No Response)

Reviewer #3: (No Response)

7. PLOS authors have the option to publish the peer review history of their article (what does this mean?). If published, this will include your full peer review and any attached files.

Reviewer #1: No

Reviewer #3: No

---

## [Editor Report · Acceptance letter]

16 Sep 2024

PONE-D-23-28461R3 

PLOS ONE

Dear Dr. Higashiyama, 

I'm pleased to inform you that your manuscript has been deemed suitable for publication in PLOS ONE. Congratulations! Your manuscript is now being handed over to our production team.

Kind regards, 

on behalf of

Dr. Sadiq H. Abdulhussain 

Academic Editor

PLOS ONE